



# Tropospheric sources and sinks of gas-phase acids in the Colorado Front Range

James M. Mattila[1], Patrick Brophy[1], Jeffrey Kirkland[1], Samuel Hall[2], Kirk Ullmann[2], Emily V. Fischer[3], Steve Brown[4,5], Erin McDuffie[4,5,6], Alex Tevlin[7], Delphine K. Farmer[1]

[1]Department of Chemistry, Colorado State University, Fort Collins, CO, USA
[2]National Center for Atmospheric Research, Boulder, CO, USA
[3]Department of Atmospheric Science, Colorado State University, Fort Collins, CO, USA
[4]NOAA Earth System Research Laboratory, Chemical Sciences Division, Boulder, CO, USA
[5]Department of Chemistry and Biochemistry, University of Colorado Boulder, Boulder, CO, USA
[6]Cooperative Institute for Research in Environmental Sciences, University of Colorado Boulder, Boulder, CO, USA
[7]Department of Chemistry, University of Toronto, Toronto, ON, Canada

*Correspondence to:* Delphine K. Farmer (Delphine.Farmer@colostate.edu)

**Abstract.** We measured organic and inorganic gas-phase acids in the Front Range of Colorado to better understand their tropospheric sources and sinks using a high-resolution time-of-flight chemical ionization mass spectrometer. Measurements were conducted from 4 to 13 August 2014 at the Boulder Atmospheric Observatory during the Front Range Air Pollution and Photochemistry Éxperiment. Diurnal increases in mixing ratios are consistent with photochemical sources of $HNO_3$, HNCO, formic, propionic, butyric, valeric, and pyruvic acid. Vertical profiles taken on the 300 m tower demonstrate net surface-level emissions of alkanoic acids, but net surface deposition of $HNO_3$ and pyruvic acid. The surface-level alkanoic acid source persists through both day and night, and is thus not solely photochemical. Reactions between $O_3$ and organic surfaces may contribute to the surface-level alkanoic acid source. Nearby traffic emissions and agricultural activity are a primary source of propionic, butyric, and valeric acid, and likely contribute photochemical precursors to $HNO_3$ and HNCO. The combined diel and vertical profiles of the alkanoic acids and HNCO are inconsistent with dry deposition and photochemical losses being the only sinks, suggesting additional loss mechanisms.

## 1 Introduction

Organic acids comprise a major fraction of gas-phase acids in the troposphere. They influence the acidity of precipitation, fog, and cloud droplets, particularly in rural areas (Keene and Galloway, 1984; Gasche et al., 2002), and can thus impact ecosystem health (Sverdrup et al., 2001; Himanen et al., 2012). Organic acids are also involved in the formation of secondary organic aerosol (SOA) (Vogel et al., 2013; Yatavelli et al., 2014; Yatavelli et al., 2015), which affects human health, visibility, and climate. Yatavelli et al. (2015) estimated that molecules containing carboxylic acid moieties account for $10-50\%$ of continental Northern Hemispheric organic aerosol mass. Sources and sinks determine tropospheric concentrations of gas-phase organic acids, and thus their impacts on biological health and air quality. However, several model-measurement comparisons for tropospheric formic and acetic acid indicate missing sources, potentially coupled to missing sinks (Paulot et al., 2011; Yuan et al., 2015; Millet et al., 2015; Schobesberger et al., 2016). Model-measurement comparisons for other tropospheric organic acids are lacking. Field and laboratory measurements investigating the sources and sinks of these compounds are therefore necessary to reduce model uncertainties and improve our understanding of organic acids in the troposphere.

A variety of primary biogenic and anthropogenic sources can introduce organic acids into the troposphere. Several organic acids have been identified in vegetative emissions (Kesselmeier et al., 1998; Kesselmeier, 2001), soil emissions (Sanhueza and Andreae, 1991; Enders et al., 1992), and biomass burning (Goode et al., 2000). Automobile exhaust is also a primary source of





alkanoic acids, with formic ($CH_2O_2$) and acetic ($C_2H_4O_2$) acid typically being the most abundant in these emissions (Kawamura et al., 1985; Kawamura et al., 2000; Friedman et al., 2017). Secondary production from the photochemical oxidation of volatile organic compounds (VOCs) serves as another major source. Photochemical oxidation of isoprene ($C_5H_8$) produces several organic acids, including formic and pyruvic acid ($C_3H_4O_3$) (Orzechowska and Paulson, 2005; Jacob and Wofsy, 1988; Paulot et al., 2009;

Paulot et al., 2011). Friedman et al. (2017) measured formic, propionic ($C_3H_6O_2$), and butyric acid ($C_4H_7O_2$) in photochemically-aged diesel exhaust. Wet and dry deposition, and photochemical loss processes are the major known tropospheric sinks of organic acids (Grosjean, 1989; Talbot et al., 1995; Atkinson et al., 2006; Grosjean, 1983). Despite their ubiquity, our understanding of tropospheric organic acid sources and sinks is incomplete. This is especially apparent for formic acid—measured tropospheric concentrations are often several times higher than modeled values (Paulot et al., 2011; Yuan et al., 2015; Millet et al., 2015;

Schobesberger et al., 2016). Model simulations have also failed to capture the temporal variation and vertical gradients of formic acid (Millet et al., 2015). These model-measurement discrepancies are likely due to underestimated sources and/or overestimated sinks, as well as missing sources and sinks that are not considered altogether.

Gas-phase inorganic acids, including nitric ($HNO_3$) and isocyanic acid (HNCO), also impact air quality. $HNO_3$ is produced in the troposphere from nitrogen dioxide ($NO_2$) reactions with hydroxyl radical (OH), and through the reaction of $NO_2$ with ozone

($O_3$). Anthropogenic emissions of nitrogen oxides ($NO_x = NO + NO_2$) from fossil fuel combustion and agricultural activity constitute a major secondary source of $HNO_3$ (Shepherd et al., 1991; Dignon, 1992; Kurvits and Marta, 1998; Almaraz et al., 2018). $HNO_3$ readily partitions into the aqueous-phase, contributes to acid deposition, and reduces the vapor pressure of water during cloud droplet growth—affecting the growth rate and resulting size of these droplets (Kulmala et al., 1993). $HNO_3$ also reacts with ammonia ($NH_3$) in the gas- or aqueous-phase to form ammonium nitrate ($NH_4NO_3$) aerosols (Adams et al., 1999). HNCO is

of growing interest because exposure levels > 1 $ppb_v$ are linked to various human health issues, including atherosclerosis, cataracts, and rheumatoid arthritis (Jaisson et al., 2011; Roberts et al., 2011). Primary emission and secondary photochemical production sources of gas-phase HNCO have been identified and reported (Borduas et al., 2013; Roberts et al., 2014), but the magnitudes of these sources remain highly uncertain (Young et al., 2012). Combustion processes, including biomass burning, gasoline/diesel fuel combustion, and tobacco smoke are a primary source of HNCO (Roberts et al., 2011; Roberts et al., 2014; Link et al., 2016).

Secondary sources of HNCO include OH oxidation of amine and amide precursors, which are particularly important in urban environments (Link et al., 2016; Roberts et al., 2014; Borduas et al., 2013). HNCO readily partitions into the aqueous-phase given its high solubility at atmospherically relevant pH values, and can hydrolyze to $NH_3$ (Roberts et al., 2011). Wet and dry deposition are other known HNCO sinks (Young et al., 2012; Roberts et al., 2014).

Here, we present ambient measurements of various gas-phase organic and inorganic acids taken during the Front Range

Air Pollution and Photochemistry Éxperiment (FRAPPÉ) in Weld County, CO (McDuffie et al., 2016; Tevlin et al., 2017; Pfister et al., 2017b; Wild et al., 2017). We use diel trends and vertical profiles of these compounds, as well as correlations in timeseries data to investigate their tropospheric sources and sinks. The peri-urban Boulder Atmospheric Observatory (BAO) site lies at the intersection of agricultural sources, traffic, oil and gas development, and other industrial processes, providing a contrast to the strictly urban or forest sites that are often the focus of atmospheric chemistry measurements.

## 2 Methods

### 2.1 Site description

Measurements took place at the BAO tower in Erie, CO during the FRAPPÉ field campaign in summer 2014. This work focuses on measurements taken between 4 and 13 August 2014. The land surrounding the tower is a sparsely vegetated region of the Colorado Front Range located on the outskirts of several urbanized Colorado municipalities (Boulder, Denver, Fort Collins,



and Greeley). The site lies about 2 km west of highway traffic from Interstate 25, is surrounded by oil and natural gas (ONG) wells, and is near (> 7 km) concentrated animal feeding operations (CAFOs) (Fig. 1) (Kaimal and Gaynor, 1983; Brown et al., 2013; Swarthout et al., 2013; Abeleira et al., 2017; Tevlin et al., 2017).

The 300 m BAO tower was equipped with an elevator carriage capable of continuous vertical movement between altitudes of 0 – 285 m, allowing for the generation of vertical profiles of measured compounds. A timeseries of carriage altitude throughout the reported measurement period is provided in Fig. S1. The carriage height was typically parked at 100 m (accounting for 62% of data described herein). This moveable carriage housed a high-resolution time-of-flight chemical ionization mass spectrometer (TOF-CIMS) allowing for fast (1 Hz) detection of gas-phase compounds (discussed further in Sect. 2.2), as well as an IRGASON Integrated $CO_2$ and $H_2O$ Open-Path Gas Analyzer, and 3-D Sonic Anemometer (Campbell Scientific) for air temperature, water

vapor, and wind speed/direction measurements. Additional meteorological measurements at 10, 100, and 300 m were provided by the BAO Tower Meteorological Station. A filter radiometer (Metcon, GmbH, Shetter et al. (2003)) measured downwelling $NO_2$ photolysis rates ($j_{NO2}$) near the base of the tower, from which total photolysis rates were calculated. Instruments to measure various trace gases of interest, including $NO_x/O_3$ (custom built Cavity Ring-Down Spectroscopy), $CO/CO_2/CH_4/H_2O$ (Picarro 6401 Cavity Ring-Down Spectrometer), and $NH_3$ (QC-TILDAS; Aerodyne Research, Inc.) were also housed on the PISA during the campaign.

The $CO/CO_2/CH_4/H_2O$ measurement details can be found in McDuffie et al. (2016) and Zaragoza et al. (2017). Instrument details on the $NH_3$ measurements are provided by Tevlin et al. (2017). All measurements presented here are reported in local time (Mountain Daylight Time; MDT; UTC – 6). Rainfall did not exceed 0.3 cm day$^{-1}$ near the site throughout the reported measurement period. We plot $j_{NO2}$ by hour of day as a proxy for solar exposure (Fig. 2). Solar exposure at the site peaks around 12:00.

**2.2 TOF-CIMS measurements**

The TOF-CIMS (Tofwerk AG and Aerodyne Research, Inc.) has been described extensively elsewhere (Bertram et al., 2011; Lee et al., 2014; Brophy and Farmer, 2015; Brophy and Farmer, 2016; Lopez-Hilfiker et al., 2016). When coupled to acetate ($CH_3COO^-$) reagent ions, this instrument detects an array of molecules including $HNO_3$, HNCO, formic, propionic, butyric, valeric ($C_5H_{10}O_2$), and pyruvic acid in the atmosphere at high acquisition rates (i.e. < 1 s time resolution). Acetate reagent ions provide high sensitivity and selectivity for gas-phase acids (Veres et al., 2008; Bertram et al., 2011; Brophy and Farmer, 2015; Brophy and

Farmer, 2016). Acetate reagent ions are generated by passing $N_2$ saturated with acetic anhydride through a $^{210}$Po ionizer (NRD). These reagent ions enter the ion-molecule reactor along with sampled ambient air and selectively ionize gas-phase acids (HA) via either a proton-exchange reaction (Veres et al., 2008) or a clustering reaction with HA followed by declustering prior to detection (Brophy and Farmer, 2016). Under both mechanisms, the analyte of interest is detected by the mass spectrometer as a deprotonated, gas-phase anion ($A^-$). Detection of acetic acid is not possible using this ion chemistry.

Instrument background is monitored hourly at the beginning of each data acquisition period using an overflow of ultra zero grade air (UZA, Airgas). Hourly online two-point external standard calibrations of formic acid are also taken in UZA prior to each ambient air measurement period, enabling direct calculation of instrument sensitivity to formic acid, and thus formic acid mixing ratios. Formic acid standard is generated from a permeation tube (Dynacal, VICI) in a heated oven held at 40 °C. Ultra-high purity (UHP) nitrogen (Airgas) flows through this permeation system, introducing the standard into the TOF-CIMS. Mass

spectral data acquisition is controlled with TofDaq Recorder (Tofwerk AG), and automated using home-built programs (LabVIEW, National Instruments). Instrument sensitivity to formic acid during the campaign was $2.35 \times 10^4$ ncps ppb$_v^{-1}$, determined from a Gaussian fit to the histogram of sensitivity values. The low dispersion in these sensitivity values (% RSD = 1.4) indicates high instrument stability throughout the campaign. We used offline external calibrations of other detected compounds to estimate mixing ratios for other gas-phase acids detected during the campaign (see Supplemental).



### 2.3 Mass spectral data processing and analysis

We process mass spectral data in Igor Pro (WaveMetrics Inc., Version 6) with Tofware (Tofwerk AG, Aerodyne Research Inc, Version 2.5.10), which determines mass spectral baseline, fitted peak shape, and peak resolution, and applies a TOF duty cycle correction (m/z = 59). We mass calibrate post-acquisition using a three-parameter fit and the $O_2^-$, $Cl^-$, $CHO_2^-$, $NO_2^-$, $C_2H_3O_2^-$, $NO_3^-$, and $I^-$ peaks; these peaks were fully resolved during the measurements with consistently high signal throughout the measurement and calibration periods. Additional conjugate bases of various other organic acids (such as $C_3H_3O_2^-$ and $C_3H_5O_3^-$), as well as the [acetic acid + acetate] cluster ($C_4H_7O_4^-$) are included in the mass calibration when signal is sufficiently high and the peaks do not contain interferences. During FRAPPÉ, the mass accuracy of the TOF-CIMS was 2 ppm (campaign average of mass calibrant ions), and the resolution (m/$\Delta$m) was > 3000. Tofware's high-resolution peak fitting procedures extract timeseries of detected compounds. Further data analysis, including background subtraction, normalization, mixing ratio calculation, and the generation of diel and vertical profiles are performed in Igor Pro. Mass spectral data are normalized to convert raw instrumental ion counts per second (cps) to normalized cps (ncps) by multiplying the measured analyte signal by the ratio of acetate reagent ion signal taken during an instrumental background to reagent ion signal taken during periods of analyte measurements (Bertram et al., 2011).

### 3 Results

Campaign statistics for each measured acid are reported in Table 1. Formic acid was the most abundant compound quantified by TOF-CIMS, with an average mixing ratio of 1.9 ppb$_v$. Compounds with negative minimum mixing ratio values are reported as below the instrumental limit of detection (LOD). We determined correlation coefficients between each measured gas-phase acid, and for each gas-phase acid compared to CO (subsampled from 8:30 to 10:30), $NH_3$, air temperature, and $j_{NO2}$ (Table 2). Timeseries for measured acid mixing ratios are provided in Fig. S2.

We bin mixing ratio data from periods of constant carriage height (100 m) by hour of the day to generate diel profiles for all gas-phase acids (Fig. 2). A diel maximum occurs between 09:00 – 10:00 for $HNO_3$, and 12:00 – 15:00 for all other acids. Secondary maxima occur around 09:00 – 10:00 for propionic, butyric, and valeric acid.

We select three typical vertical profiles to investigate noon, night, and morning trends (Fig. 3); these profiles started at 12:00 on 12 August 2014, 03:30 on 13 August 2014, and 10:00 on 13 August 2014. We observe hysteresis in analyte measurements during periods of downward carriage movement, potentially due to shaking or inlet interferences in the elevator carriage, so focus our analysis solely on profiles collected during upward carriage movement. Unfortunately, these three profiles are the sole profiles in which upward carriage movement occurred simultaneously with ambient air sampling during morning or noon periods, preventing us from replicating those time periods. Vertical profiles for nearly all gas-phase acids show a strong, near-surface gradient below 75 m. Negative gradients (i.e. mixing ratio decreases with height above ground) imply upward fluxes and net surface-level emission, while positive gradients imply downward fluxes, or net deposition. $HNO_3$ and pyruvic acid exhibit surface-level deposition in their noon, night, and morning vertical profiles. HNCO had a strong negative near-surface gradient during noon, and a weaker negative gradient during morning. All alkanoic acids exhibit surface-level emission in their noon, night, and morning vertical profiles (except for butyric acid during nighttime).

### 4 Discussion

#### 4.1 Alkanoic acids

Formic acid at BAO (1.9 ppb$_v$ average) is comparable to previous measurements in urban and rural areas (Glasius et al., 2000; Kawamura et al., 1985; Veres et al., 2011). All alkanoic acid mixing ratios increase throughout the day (Fig. 2), consistent with previously reported diurnal trends (Veres et al., 2011; Brophy and Farmer, 2015). Additionally, formic acid mixing ratios



correlate strongly with $j_{NO2}$ ($r^2 = 0.738$). These data point to a photochemical source of alkanoic acids, consistent with known reaction mechanisms. For example, ozonolysis of alkenes and photooxidation of isoprene are photochemical sources of formic acid in the troposphere (Orzechowska and Paulson, 2005; Jacob and Wofsy, 1988; Paulot et al., 2009; Paulot et al., 2011; Millet et al., 2015). Alkanoic acids are also produced during photooxidation of diesel exhaust (Friedman et al., 2017).

5        Vertical profiles indicate an additional, non-photochemical surface source of alkanoic acids. Alkanoic acid vertical profiles exhibit negative gradients, demonstrating upward fluxes from near the surface (< 75 m) to the atmosphere throughout the day and night (with the exception of butyric acid at night) (Fig. 3). Possible drivers of this near-surface source are explored below. While photochemistry is an important atmospheric source of all observed alkanoic acids, the persistent near-surface gradient through both night and day requires an additional non-photochemical source at or near the surface.

10        Light- and temperature-dependent primary emissions of alkanoic acids from the stomata of plants have been reported previously (Kesselmeier et al., 1998), and could contribute to their observed diurnal increases (Fig. 2). However, vegetation in the region is sparse, particularly during the hot, dry Front Range summer. Further, the near-surface source persists through both day and night, while biogenic emissions typically cease during the night when stomata are closed and photosynthesis has stopped. Soil emissions are another plausible source of alkanoic acids, but typically thought to be minor (Sanhueza and Andreae, 1991; Enders

et al., 1992). We thus expect that biogenic sources of the alkanoic acids were minor during the campaign.

       Traffic emissions are a primary, and potentially secondary, source of propionic, butyric, and valeric acid. These compounds have been observed as primary and secondary emissions from automobile exhaust (Kawamura et al., 1985; Kawamura et al., 2000; Friedman et al., 2017). Peaks in the diel profiles of these compounds between 09:00 – 10:00 are consistent with morning rush-hour traffic and $NO_x$ (Fig. S4). $NO_x$ is commonly used as a tracer for near-field automobile emissions (Abeleira et

al., 2017). CO is also an effective tracer for primary automobile emissions in the Front Range (Abeleira et al., 2017). Propionic, butyric, and valeric acid correlate particularly well with CO during morning rush-hour periods ($r^2 = 0.635$ for propionic, $r^2 = 0.615$ for butyric, and $r^2 = 0.721$ for valeric), suggesting that traffic dominated the source of these acids during that time. Correlations between the three acids and CO throughout the entire timeseries were lower ($r^2 = 0.237$ for propionic, $r^2 = 0.062$ for butyric, and $r^2 = 0.128$ for valeric), indicating that other sources influenced their gas-phase mixing ratios throughout the rest of the day. Much

like CO, propionic, butyric, and valeric acid showed noticeable increases in measured mixing ratios from winds between 90° – 180° during morning rush-hour periods, consistent with the hypothesis that nearby traffic dominated the propionic, butyric, and valeric acid sources during morning rush hour (Fig. 4). McDuffie et al. (2016) and Zaragoza et al. (2017) have shown that wind direction analysis alone is not effective for determining the direction/magnitude of upwind sources near BAO, due to significant mixing and recirculation of air near the site. However, we use these profiles merely to show that these acids share the same

incoming air parcels measured at the site as CO—i.e. these compounds are transported to the site from the same traffic source, irrespective of the exact direction of this source relative to the site. Formic acid behaves quite differently from the other alkanoic acids with respect to a potential traffic source. While automobile emissions are a known production source of formic acid (Kawamura et al., 1985; Kawamura et al., 2000; Friedman et al., 2017), formic acid did not exhibit a morning rush hour maximum, was only weakly correlated to CO during rush hour ($r^2 = 0.026$), and did not share the rush hour directionality with the other acids

(Fig. 4). Despite the demonstrable traffic source of propionic, butyric and valeric acid, there is little evidence that traffic was the near-surface source observed in the vertical profiles (Fig. 3).

       Agricultural activity is another primary emission source of alkanoic acids (McGinn et al., 2003; Paulot et al., 2011), and may have contributed to the observed alkanoic acid mixing ratios. $NH_3$ in the Colorado Front Range comes primarily from agricultural sources (Tevlin et al., 2017). $NH_3$ correlates more strongly with butyric ($r^2 = 0.453$) and valeric ($r^2 = 0.355$) acids than

propionic acid ($r^2 = 0.221$) throughout the entire day. Like $NH_3$ (Fig. S5), all three acids increase with winds from 0° – 90°, which



is likely attributable to transport from nearby CAFOs (Fig. S6). Correlations between these acids and NH$_3$ were stronger during daytime (12:00 – 5:00) periods (r$^2$ = 0.517 for propionic, r$^2$ = 0.649 for butyric, and r$^2$ = 0.426 for valeric), suggesting that agricultural activity was predominantly a daytime source. Agricultural sources of formic acid have been suggested previously (Paulot et al., 2011). The weak correlation with NH$_3$ (r$^2$ = 0.044 for entire day, r$^2$ = 0.228 during daytime) suggests that agricultural activity was likely a minor daytime source of formic acid.

Photochemical oxidation of VOCs is an established atmospheric source of formic acid, and is consistent with the observed formic acid diel cycle and correlation with j$_{NO2}$ (r$^2$ = 0.738). Formic acid is produced during ozonolysis of ethene and propene (Atkinson et al., 2006; Millet et al., 2015), both of which have known combustion sources (Gilman et al., 2013), and during OH oxidation of diesel emissions (Friedman et al., 2017). ONG wells were dominantly to the east of the site (Fig. 1). These wells were a potential source of formic acid precursors due to the combustion processes associated with their operation (such as gas flaring). Isoprene is a known photochemical precursor of formic acid (Jacob and Wofsy, 1988; Orzechowska and Paulson, 2005; Paulot et al., 2009), though it has been observed in relatively low mixing ratios at BAO during the summer (0.2 ± 0.3 ppb$_v$ average) (Abeleira et al., 2017). Further, anthropogenic sources dominate summertime OH reactivity at the site (Abeleira et al., 2017), and reports of isoprene oxidation as a major source of formic acid typically occur in heavily vegetated areas (Jacob and Wofsy, 1988; Stavrakou et al., 2012; Millet et al., 2015). The diurnal increases in propionic, butyric, and valeric acid reported here are consistent with previous field observations (Satsumabayashi et al., 1995; Veres et al., 2011) and reported photochemical production mechanisms of these compounds (Satsumabayashi et al., 1995; Orzechowska et al., 2005).

Photochemical sources are unlikely responsible for the near-surface source that persists thought the day. We note that while photochemical processing of anthropogenic precursors is a known source of HNO$_3$ and pyruvic acid (see Sect. 4.2), the vertical profiles of these two acids are dominated by dry deposition and not surface sources. However, HNCO also has known photochemical and traffic sources, and displays a negative (upward flux) daytime, but not nighttime, near-surface vertical gradient (see Sect. 4.3). While it is possible that photochemical or traffic sources could cause the surface source implied by the alkanoic acid vertical profiles, it is less likely that they are responsible for the nighttime source.

The identity of the surface-level non-photochemical source thus remains unclear. Several other recent studies invoke missing alkanoic acid sources—i.e. sources not typically considered when modeling tropospheric VOC budgets. Paulot et al. (2011) suggested that photochemical aging of aerosols could serve as a major missing source of formic and acetic acid. Model-measurement discrepancies led Schobesberger et al. (2016) to suggest significant, unresolved surface-level sources of formic acid, although that study noted temperature and light dependences similar to emission parameterizations of other well-characterized biogenic VOCs. Millet et al. (2015) and Nguyen et al. (2015) also observed similar model-measurement discrepancies of formic acid, which were attributed to missing/underestimated chemical production and/or biogenic emissions sources.

Multiple processes could be responsible for the observed surface-level source of alkanoic acids. We hypothesize that reactions between O$_3$ and organic surfaces (i.e. soil, organic films) could be one non-photochemical surface-level source of alkanoic acids near the site. Reactions of O$_3$ on organic surfaces such as organic films (Donaldson et al., 2005), plant surfaces (Cape et al., 2009; Jud et al., 2016), and human skin (Liu et al., 2016; Liu et al., 2017) have been reported previously. Soil organic matter and organic films are often rich in alkenes (Vancampenhout et al., 2009; Donaldson et al., 2005; Simpson et al., 2006), which undergo ozonolysis reactions in the presence of O$_3$ (Criegee, 1975; Wolff et al., 1997). Hydroxyalkyl hydroperoxides formed via the ozonolysis of alkenes can further decompose to alkanoic acids (Moortgat et al., 1997; Anglada et al., 2002; Hasson et al., 2003; Millet et al., 2015). O$_3$ mixing ratios measured at the site were relatively high at nighttime (~ 40 ppb$_v$) (Fig. S7), further suggesting that this process may contribute to the persistent upward flux of alkanoic acids through both day and night.

.



Wet and dry deposition are major sinks of alkanoic acids (Grosjean, 1989; Talbot et al., 1995). Removal via reactions with OH are slow, corresponding to atmospheric lifetimes of several days (Dagaut et al., 1988). $C_1 - C_5$ alkanoic acids have negligible absorption cross sections at wavelengths greater than ~250 nm (Singleton et al., 1987; Vicente et al., 2009); photolysis is thus not considered to be a major tropospheric alkanoic acid sink. Wet deposition was minimal in the Front Range during the study period due to the lack of rainfall events during the reported measurement period. Dry deposition should thus have been the only major alkanoic acid sink during the night. However, the vertical profiles showed upward fluxes of these compounds at night (Fig. 3). The nocturnal decrease in mixing ratio necessitates an additional non-photochemical sink for these compounds, consistent with previous suggestions by Brophy and Farmer (2015). Cloud processing, gas-particle phase partitioning, and aqueous-phase reactions are possible alkanoic acid sinks. The high Henry's Law constants (H) of these acids suggest that aqueous-phase partitioning (aqueous aerosols, fog and cloud droplets, etc.) would be favorable (H = $5.5 \times 10^3$, $5.7 \times 10^3$, $4.7 \times 10^3$, and $2.2 \times 10^3$ mol $L^{-1}$ $atm^{-1}$ for formic, propionic, butyric, and valeric acid, respectively at T = 298 K) (Khan et al., 1995). However, this was likely not a significant sink given the arid climate of the Front Range. Carlton and Turpin (2013) suggest that liquid water concentration in the Front Range during summer is ~1 µg $m^{-3}$. Combining this with known constants, campaign mean mixing ratios, and meteorological conditions, aqueous-phase partitioning accounts for an estimated loss of $< 2 \times 10^{-10}$ $ppb_v$ of each alkanoic acid (see Supplemental). While this ignores effects of pH and other dissolved ions on solubility, aqueous partitioning is unlikely a substantial loss process for the alkanoic acids during the measurement campaign. Gas-phase reactions between the alkanoic acids and atmospheric bases, such as $NH_3$, amines, or amides have not been reported extensively. Grosjean (1989) suggested that carboxylic acids can react with $NH_3$ in the atmosphere to produce carboxylate ammonium salts, though the importance of this process as a tropospheric sink of alkanoic acids remains uncertain.

**4.2 Nitric and pyruvic acid**

$HNO_3$ and pyruvic acid follow similar diel and vertical trends ($r^2 = 0.603$), and their diel profiles are consistent with photochemical sources (Fig. 2). Additionally, pyruvic acid correlates particularly well with $j_{NO2}$ ($r^2 = 0.783$). Unlike the alkanoic acids, $HNO_3$ and pyruvic acid exhibit persistent net deposition to the surface near the site during the noon, night, and morning periods (Fig. 3).

Traffic was likely an important secondary source of $HNO_3$ and pyruvic acid. $HNO_3$ is produced from $NO_2$ + OH, and pyruvic acid is produced from photooxidation of diesel exhaust (Friedman et al., 2017), including from 1,3,5-trimethylbenzene in the presence of $NO_x$ (Praplan et al., 2014). Both $NO_x$ and 1,3,5-trimethylbenzene are abundant components of automobile exhaust (Nelson and Quigley, 1984; Khoder, 2007). However, correlations between these acids and CO during morning rush-hour traffic were weak ($r^2 = 0.274$ for $HNO_3$, and $r^2 = 0.264$ for pyruvic acid), perhaps unsurprising as CO is directly emitted from traffic exhaust, whereas $HNO_3$ and pyruvic acid require photochemistry. This observation suggests that regional, rather than nearby traffic is the source of these two acids. Neither $HNO_3$ nor pyruvic acid correlate with $NH_3$. However, agricultural activity is a known source of $NO_x$, which is primarily emitted from fertilizer and heavy-duty diesel farm vehicles (Shepherd et al., 1991; Kurvits and Marta, 1998). We therefore speculate that agricultural sources also served as a secondary source of $HNO_3$ near the site. Reports of pyruvic acid from agricultural sources are sparse, and we cannot evaluate the potential of this source with the data presented here. ONG and industrial activities are also sources of $NO_x$ in the Front Range (Pfister et al., 2017a), and thus likely secondary sources of $HNO_3$. There is no evidence for strong surface-level emission sources of $HNO_3$ or pyruvic acid in the vertical profile data.

Vertical profiles of both $HNO_3$ and pyruvic acid are consistent with dry deposition (Fig. 3). While both $HNO_3$ and pyruvic acid readily partition into the aqueous-phase (H = $2.1 \times 10^5$ mol $L^{-1}$ $atm^{-1}$ and $3.1 \times 10^5$ mol $L^{-1}$ $atm^{-1}$ for $HNO_3$ and pyruvic acid, respectively) (Khan et al., 1995; Schwartz and White, 1981), we estimate that aqueous-phase partitioning is a negligible sink for



both compounds. Photochemistry is not a major sink of HNO$_3$, but pyruvic acid readily undergoes photolysis—corresponding to a typical atmospheric lifetime of a few hours (Grosjean, 1983). However, the reaction of pyruvic acid with OH is negligible, corresponding to a lifetime on the order of months (Grosjean, 1983). Reactions between ambient NH$_3$ and HNO$_3$ produce NH$_4$NO$_3$ aerosol (Li et al., 2014), though we estimate that this process would not be a significant sink of gas-phase HNO$_3$ (see Supplemental).

**4.3 Isocyanic acid**

The afternoon diurnal peak of HNCO is consistent with photochemical production sources (Fig. 2). The diel profile of HNCO at BAO is similar to that observed previously in rural NE Colorado during BioCORN 2011, which was attributed to secondary photochemical production from amine and formamide (Roberts et al., 2014). The daytime vertical profiles show clear, upward fluxes of HNCO from the surface (Fig. 3). This vertical gradient is strongest at noon, smaller in the morning and unclear
at night, implying a surface source that is driven by photochemistry.

Roberts et al. (2014) suggested that farmland and cattle feedlots located along Interstate 25 serve as a source of photochemical precursors (various amine and amide compounds) of HNCO in the Colorado Front Range. This is supported by the correlation between HNCO and temperature (r$^2$ = 0.773) as these agricultural precursors are likely temperature-dependent. Sintermann et al. (2014) reported that alkaline compounds such as amines undergo enhanced volatilization from agricultural sites
when air temperatures are higher due to a decrease in temperature-dependent solubility and an increase in soil/waste pH due to accelerated hydrolysis of urea. HNCO mixing ratios were possibly influenced by additional sources, including traffic, ONG wells, and industrial activity. Traffic exhaust is a primary emission source of HNCO (Brady et al., 2014; Link et al., 2016), but the lack of a morning rush-hour peak or correlation with CO suggests that it was not a strong primary source of HNCO at the site (Fig. 2). Link et al. (2016) found that diesel exhaust was a precursor for photochemical HNCO production, but Jathar et al. (2017) suggested
that the kinetics do not substantially outcompete dilution, and that urban HNCO is not strongly enhanced by diesel exhaust photochemistry.

Dry deposition is a major sink of HNCO (Roberts et al., 2014; Young et al., 2012), although HNCO readily partitions into the aqueous-phase (H = 10$^5$ mol L$^{-1}$ atm$^{-1}$), where it can hydrolyze to NH$_3$ (Roberts et al., 2011). We estimate that aqueous partitioning of HNCO was negligible. No major sinks of HNCO aside from wet deposition, dry deposition, and aqueous-phase
chemistry have been reported, and photochemical loss reactions are negligible, with a photolysis lifetime of several months (Roberts et al., 2011), and an OH oxidation lifetime of several years (Tsang, 1992; Roberts et al., 2011; Borduas et al., 2016). HNCO has a relatively high gas-phase acidity (Wight and Beauchamp, 1980; Veres et al., 2010), and we hypothesize that non-photochemical gas-phase acid-base reactions could be a nighttime sink for HNCO.

**5 Conclusions**

Diurnal increases in all gas-phase acids are consistent with photochemical sources. We observe net surface-level emissions of alkanoic acids through both day and night, suggesting additional non-photochemical surface sources. We speculate that reactions between O$_3$ and organic surfaces (i.e. soil, organic films) near the site could be driving this persistent upward alkanoic acid flux. Correlations with chemical tracers suggest that traffic emissions and agricultural activity near the site are a primary source of propionic, butyric, and valeric acid, and potentially a secondary source of HNO$_3$, and HNCO.
Dry deposition is the dominant sink of HNO$_3$ and pyruvic acid, but was not large enough to out-compete the surface source of the alkanoic acids. Which sinks control the lifetime of the alkanoic acids remain unclear. A non-photochemical sink of HNCO on top of dry deposition is also suggested by the vertical profile data and warrants further investigation.



*Data availability.* Meteorological data taken at the BAO tower are available at: https://www.esrl.noaa.gov/psd/technology/bao/browser/. All other data supporting the analysis are available at: https://www-air.larc.nasa.gov/missions/discover-aq/discover-aq.html.

*Competing interests.* The authors declare that they have no conflict of interest

*Acknowledgements.* We acknowledge the Colorado Department of Public Health and Environment (CDPHE) for funding the work. We thank Dan Wolfe and Gerd Hübler for substantial logistical support, Jennifer Murphy for assistance with QC-TILDAS measurements, and Michael Link for assistance with the meteorology data

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



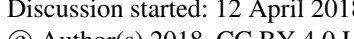

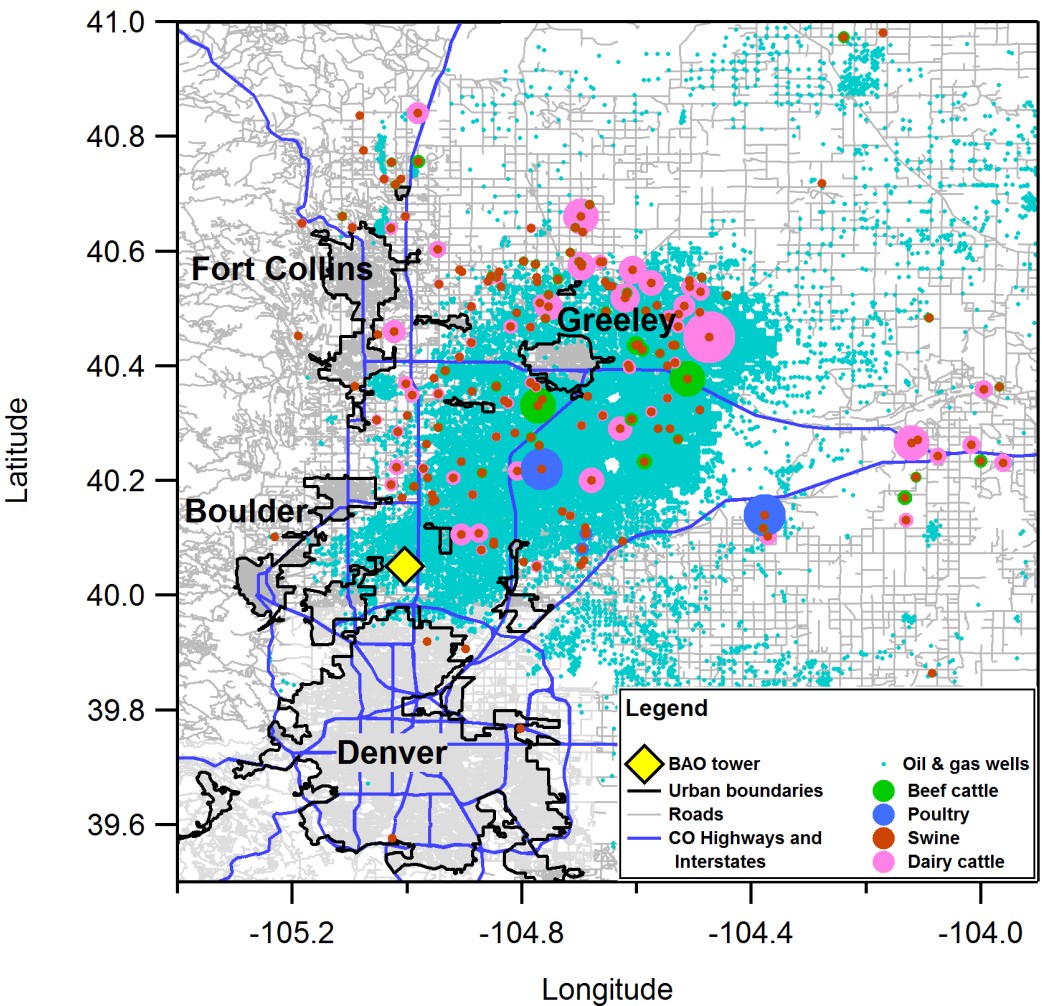

Figure 1. Area surrounding BAO site, including major nearby urban municipalities, roads and highways, ONG wells, and CAFOs.
5  CAFOs are colored by operation type and sized by number of animal units per operation.



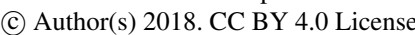

**Figure 2. (a–g) Diel profiles for all detected gas-phase acids at 100 m. (h) Diel profile for $j_{NO2}$ measured at the site. Data are binned by hour. Data points are means of hourly bins. Shaded area represents ± one standard deviation of binned data.**

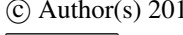



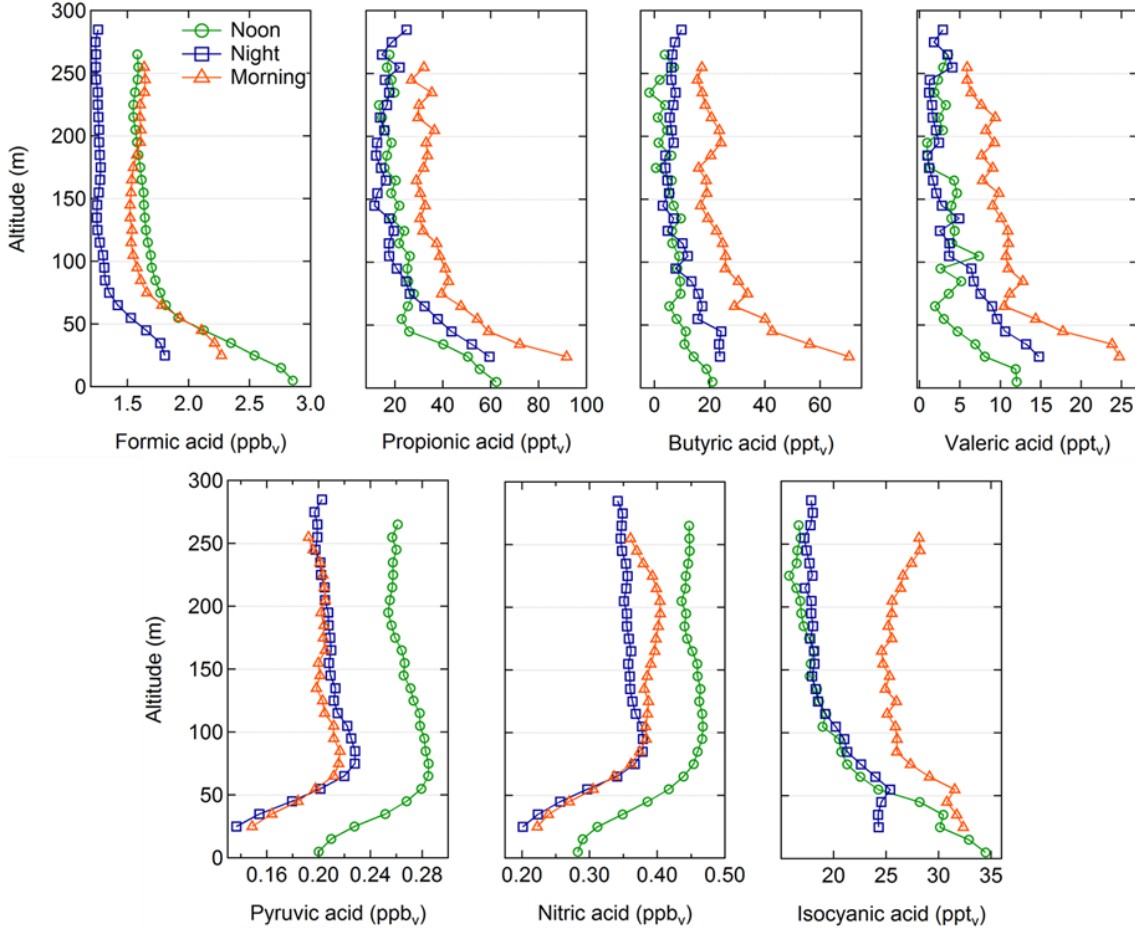

**Figure 3. Vertical profiles for all detected gas-phase acids at representative noon, night, and morning periods, showing mixing ratio as a function of altitude. Data are binned by altitude (10 m per bin). Data points are means of each bin. Error bars have been removed for clarity, and are included in Fig. S3.**



**Figure 4. Wind plots of (a) formic acid, (b) propionic acid, (c) butyric acid, (d) valeric acid, and (e) CO measured at the site. Data are selected during periods of morning rush-hour traffic (08:30–10:30). Data points are colored by mixing ratio. Radial and angular axes represent wind speed (m s⁻¹) and direction (degrees), respectively. Degrees correspond to cardinal directions (i.e. 0° is N, 90° is E, etc.).**





**Tables**

Table 1 – Campaign statistics for measured gas-phase acids.

| Gas-phase acid | Mean (ppb$_v$) | Max (ppb$_v$) | Min. (ppb$_v$) | Standard deviation (ppb$_v$) |
|---|---|---|---|---|
| Formic | 1.9 | 3.6 | 1.0 | 0.4 |
| Propionic | 0.06 | 0.70 | Below LOD | 0.03 |
| Butyric | 0.03 | 0.16 | Below LOD | 0.02 |
| Valeric | 0.01 | 0.06 | Below LOD | 0.01 |
| Pyruvic | 0.18 | 0.51 | Below LOD | 0.06 |
| Nitric | 0.30 | 1.11 | 0.00 | 0.07 |
| Isocyanic | 0.03 | 0.07 | 0.00 | 0.01 |

Table 2 – Correlation coefficients (r$^2$) for each gas-phase acid in the leftmost column compared to other gas-phase acids, chemical

5    tracers, and other meteorological parameters in the table header (Propion. = propionic acid, Isocyan. = isocyanic acid, Temp. = air temperature).

| | Formic | Isocyan. | Pyruvic | Propion. | Valeric | Nitric | Butyric | CO | NH$_3$ | Temp. | j$_{NO2}$ |
|---|---|---|---|---|---|---|---|---|---|---|---|
| Formic | — | 0.375 | 0.194 | 0.257 | 0.120 | 0.091 | 0.089 | 0.026 | 0.044 | 0.504 | 0.738 |
| Isocyan. | 0.375 | — | 0.030 | 0.102 | 0.007 | 0.005 | 0.001 | 0.093 | 0.002 | 0.773 | 0.411 |
| Pyruvic | 0.194 | 0.030 | — | 0.077 | 0.074 | 0.603 | 0.068 | 0.264 | 0.002 | 0.560 | 0.783 |
| Propion. | 0.257 | 0.102 | 0.077 | — | 0.776 | 0.231 | 0.714 | 0.635 | 0.221 | 0.058 | 0.310 |
| Valeric | 0.120 | 0.007 | 0.074 | 0.776 | — | 0.312 | 0.856 | 0.721 | 0.355 | 0.005 | 0.331 |
| Nitric | 0.091 | 0.005 | 0.603 | 0.231 | 0.312 | — | 0.332 | 0.274 | 0.113 | 0.005 | 0.382 |
| Butyric | 0.089 | 0.001 | 0.068 | 0.714 | 0.856 | 0.332 | — | 0.615 | 0.453 | 0.017 | 0.365 |