# Peer review of "Tropospheric sources and sinks of gas-phase acids in the Colorado Front Range"

_Atmospheric Chemistry and Physics, 2018_

## Referee Comment (RC1) · Anonymous Referee #2 · 8 May 2018

Mattila et al. present measurements of atmospheric acids from a field campaign in Colorado, and explore the observations in terms of our understanding of the atmospheric budgets of these species. The elevator-borne acid measurements by acetate-CIMS at the 300m BAO tower provide unique vertical information to test ideas about acid sources and sinks. The paper is brief but nonetheless makes a clear and useful contribution to the literature on this topic. The writing is succinct and effective. I include some minor comments and suggestions below for the authors to consider.

The authors speculate that surface reactions of ozone might be responsible for the observed acid enhancements, but do not attempt to test this in any way, though ozone was also measured. Are there temporal or vertical correlations that provide any evidence for this? What kind of yield / precursor abundance would be required for this to

[Figure]

work (esp for formic acid, with its 2-3 ppb enhancement)?

4, 29 and Fig 3. "vertical profiles show a strong, near-surface gradient below 75m". Indeed, the profiles tend to show this gradient at the same altitude regardless of time of day (morning, noon, night). Wouldn't we expect the positive or negative vertical gradients to manifest through a deeper layer of the atmosphere for the daytime profiles (due to mixing depth changes)?

1, 26-27 "influence the acidity of precipitation, fog, and cloud droplets (...) and can thus impact ecosystem health". The papers cited appear to refer to impacts associated with industrial release of organic acids in the first case, and with agricultural treatments in the second case. Are these relevant to the quantities found in wet deposition?

5, 35 "despite the demonstrable traffic source of priopionic, butyric, and valeric acid, there is little evidence that traffic was the near-surface source observed in the vertical profiles". The basis for this argument is not clear to me.

Supplement, estimating aqueous-phase partitioning of gas-phase acids. "this estimation is limited in that it does not account for the effects of pH or other dissolved ions of [note, should be "on"] a given acid's acidity, but we would not expect a change of several orders of magnitude by accounting for these effects." Given the environmental conditions at hand I think you are probably correct about this conclusion. However H can indeed vary a lot with pH and it would be straightforward to repeat the calculation using effective Henry's law constants for a feasible range of pH to demonstrate that your conclusion is robust.

3, 36 "2.35E4 ncps/ppb". Ncps is not defined until the subsequent paragraph; consider re-ordering or inserting 'as defined below'. I suggest also reporting here the raw sensitivity in cps/ppb as this gives a more directly interpretable measure of the instrumental response / LOD.

4, 5: misplaced comma at beginning of line

5, 13 "biogenic emissions typically cease during the night". Should specify here that this is the case for light-dependent emissions but not for solely temperature-dependent emissions.

3, 14 "PISA" not defined

---

## Referee Comment (RC2) · Anonymous Referee #1 · 12 Jun 2018

General Comments

Mattila et al. present measurements of gas phase acids from the BAO tall tower in Erie, Colorado. The novelty of this paper are the vertical profiles presented. The dataset and resulting manuscript are very brief but add in a substantial way to the existing literature on alkanoic acids. I think the paper should be published following the authors attention to the comments below:

1) The authors suggest that the strong gradient in formic acid could be sustained by O3 deposition to the surface. Is this possible? Using an upper limit for O3 deposition velocity and unit yield for formic acid production, is the surface flux large enough to sustain the concentrations seen here?

[Figure]

2) The vertical profiles show a strong inflection point at 75m. It is not clear what is driving this. The authors should include some discussion of vertical mixing in this region that could lead to this. Further, I was very surprised that the vertical profiles look almost identical (if normalized to concentration at 250m) over the course of the day. It would be helpful for the authors to provide some discussion of why the profiles are so similar in morning, at noon, and at night.

3) It would be extremely helpful to also include paired vertical profiles for T, CO, NOx, O3, and H2O to assess the extent of vertical mixing during these profiles. I recognize that these measurements are discussed in McDuffie (2016), but it would be very nice to include the vertical profiles here as a reference panel in Fig. 3 for the profiles used in this study.

4) There is no discussion of the inlet used or inlet characterization in the manuscript. This should be included. What is the inlet transmission efficiency for these molecules and how was this corrected for? I also do not see a discussion in the supplement. In addition, how substantial is the water dependence in sensitivity and how was this accounted for? Is there enough of a gradient in H2O to make this important for the interpretation of the vertical profiles? There is mention of hysteresis that could be related to inlet transmission in section 3. This should be elaborated on.

---

## Author Comment (AC1) · 20 Jul 2018

The comment was uploaded in the form of a supplement:
https://www.atmos-chem-phys-discuss.net/acp-2018-326/acp-2018-326-AC1-
supplement.pdf

---

## Author Response (AR1)

We thank both reviewers for thoughtful and detailed comments. We have outlined each comment in bold, and our responses in plain text underneath. We respond to Reviewer 2 first, as many of those questions provide useful context for our responses to Reviewer 1.

**Responses to Reviewer #2:**

**1) The authors speculate that surface reactions of ozone might be responsible for the observed acid enhancements, but do not attempt to test this in any way, though ozone was also measured. Are there temporal or vertical correlations that provide any evidence for this? What kind of yield / precursor abundance would be required for this to work (esp for formic acid, with its 2-3 ppb enhancement)?**

The source of formic acid is an interesting question, and one that we do not have a clear answer for – merely suggestions of options. To clarify, we speculate that that surface reactions of ozone may contribute to the acid enhancements, but cannot fully attribute this potential source. We have clarified the text (italics represent addition) to make it clear that we are not attributing the organic acid gradients entirely to surface+$O_3$ reactions:

> "Multiple processes could be responsible for the observed surface-level source of alkanoic acids. We hypothesize that reactions between $O_3$ and organic surfaces (i.e. soil, organic films) could be one non-photochemical surface-level source of alkanoic acids near the site, *though unlikely to account for the entire source*."

We later clarify: *"However, we emphasize that while $O_3$ reactions with surfaces could act as one source of organic acids, there is no evidence that they account for the entire surface-level organic acid source."*

However, we did indeed make ozone measurements. Noon, night, and morning vertical profile measurements of ozone (along with $NO_x$ CO, ambient temperature, and relative humidity) have been added to the Supplemental Info (Figure S3), and has been reproduced below (Figure R1). The positive concentration gradient of ozone with respect to height during nighttime is consistent with dry deposition – that is, a nocturnal surface-level sink of ozone (and with the hypothesis that ozone reacts with organic surfaces to produce alkanoic acids). Photochemical production is the dominant ozone source during the daytime, and, consistent with that idea, we do not observe vertical gradients in ozone in the morning or noon vertical profiles.

Numerous assumptions are required to directly estimate the yields/precursor abundances required for this source – mostly due to limited micrometeorology measurements on or near the tower during the campaign dates. Due to our discomfort with these assumptions (and the subsequent order of magnitude differences in our estimates), we have not provided a source yield estimate. However, we have added the following text to the manuscript into the relevant part of the Discussion:

> "*We report noon, night, and morning vertical profile measurements of $O_3$ in Figure S3. The positive concentration gradient of $O_3$ with respect to height during nighttime is consistent with a nocturnal surface-level sink of $O_3$, and the hypothesis that $O_3$ reacts with organic surfaces to produce alkanoic acids. Known photochemical production mechanisms are the dominant $O_3$ source throughout the daytime, and no net surface-level exchanges are observed in the morning or noon vertical profiles.*"

[Figure]

**Figure R1 (and S3). Vertical profiles of O₃, NOx, CO, relative humidity, and air temperature at representative noon, night, and morning periods.**

**2) 4, 29 and Fig 3. "vertical profiles show a strong, near-surface gradient below 75m". Indeed, the profiles tend to show this gradient at the same altitude regardless of time of day (morning, noon, night). Wouldn't we expect the positive or negative vertical gradients to manifest through a deeper layer of the atmosphere for the daytime profiles (due to mixing depth changes)?**

The strong gradients observed in these vertical profiles are likely within the surface layer (i.e. lowest layer) of the tropospheric boundary layer, which occurs within both the stable boundary layer (during night) and mixed layer (during day). The height of this layer remains relatively consistent during both day and night. The dynamics of the surface layer at BAO are such that micrometeorology and fluxes display little variability throughout the day, leading to a nearly constant exchange of atmospheric scalars and

pollutants within this layer (Oke, 1987). This can be seen by the fairly constant concentration gradients observed in most of the acid vertical profiles near the surface.

**3) 1, 26-27 "influence the acidity of precipitation, fog, and cloud droplets (. . .) and can thus impact ecosystem health". The papers cited appear to refer to impacts associated with industrial release of organic acids in the first case, and with agricultural treatments in the second case. Are these relevant to the quantities found in wet deposition?**

Regarding the first citation: Keene and Galloway (1984) report that the quantities of formic and acetic acid measured in precipitation in Central Virginia contribute to 16% volume weighed free acidity, and estimate that organic acids in total contribute to 18-35% of free acidity of precipitation in the continental U.S. These findings indicate that organic acids influence precipitation acidity in more urbanized areas—albeit less than inorganic species.

Regarding the second citation: we agree with the reviewer that it may not be the clearest citation to this statement. We removed the Gasche et al. (2002) citation, and replaced it with a study by Andreae et al. (1988), as it more clearly supports the claims made in this sentence. Andreae et al. report that organic acids contribute to the majority of acidity measured in precipitation samples collected in the Central Amazon Region.

**4) 5, 35 "despite the demonstrable traffic source of propionic, butyric, and valeric acid, there is little evidence that traffic was the near-surface source observed in the vertical profiles". The basis for this argument is not clear to me.**

We agree that the wording of this sentence was unclear. We rephrased the sentence to the following: *"Despite the demonstrable importance of traffic emissions as a source of alkanoic acids in the troposphere during morning rush-hour periods, the reduction of these emissions during other times of day make it unlikely that traffic was the dominant surface-level alkanoic acid source persisting throughout the noon, night, and morning vertical profiles (Fig. 3)."*

**5) Supplement, estimating aqueous-phase partitioning of gas-phase acids. "this estimation is limited in that it does not account for the effects of pH or other dissolved ions of [note, should be "on"] a given acid's acidity, but we would not expect a change of several orders of magnitude by accounting for these effects." Given the environmental conditions at hand I think you are probably correct about this conclusion. However H can indeed vary a lot with pH and it would be straightforward to repeat the calculation using effective Henry's law constants for a feasible range of pH to demonstrate that your conclusion is robust.**

The reviewer makes an interesting point, and we have modified the text in the '*Estimating aqueous-phase partitioning of gas-phase acids*' section of the Supplemental to the following:

> *"Aqueous-phase partitioning was evaluated as a potential sink for gas-phase acids by using Henry's Law:*

$$H_x = \frac{[X]_{aq}}{P_x}$$

*where $H_x$ is the Henry's Law constant for a given gas-phase acid, and $[X]_{aq}$ and $P_x$ are the aqueous concentration and partial pressure of said acid species, respectively. $P_x$ was calculated by gas-phase acid mixing ratio data, as well as meteorological data collected during the campaign. Moles of a given acid in the aqueous-phase was determined by $[X]_{aq}$ and ambient liquid water concentration (LWC). LWC in the Front Range during the summer is estimated to be around 1 µg m$^{-3}$, based on continental estimates of LWC reported by Carlton and Turpin (2013). To account for the effects of pH on solubility, $[X]_{aq}$ was calculated as the following:*

$$[X]_{aq} = H_x P_x \left(1 + \frac{K_a}{[H^+]}\right)$$

*where $K_a$ is the acid dissociation equilibrium constant for a given acid (Levanov et al., 2017; Fischer and Warneck, 1991; Borduas et al., 2016; Smith and Martell, 2004), and $[H^+]$ is the aqueous concentration of hydronium ion. Combining aqueous-phase moles of a given acid with the ideal gas law, and meteorological data from the site yields a total loss of said acid from the gas-phase through partitioning. Total loss of each acid calculated at various atmospherically-relevant pH values are reported below. This estimation is limited in that it neglects the effects of other dissolved ions on solubility, though we would not expect a change of several orders of magnitude by accounting for these effects."*

Estimations reported in supplemental have been updated to account for variability of Henry's Law at pH values of 2 to 6, which Borduas et al. (2016) have demonstrated to be relevant pH values to atmospheric liquid water content. We have reproduced the table of these data below:

| pH | Loss via aqueous partitioning (ppbv) | | | | | | |
|----|---------|-----------|---------|---------|---------|---------|-----------|
|    | Formic | Propionic | Butyric | Valeric | Pyruvic | Nitric | Isocyanic |
| 2  | 1.4E-10 | 1.4E-10 | 1.1E-10 | 5.4E-11 | 1.5E-08 | 1.8E-05 | 1.4E-10 |
| 3  | 1.6E-10 | 1.4E-10 | 1.2E-10 | 5.4E-11 | 8.3E-08 | 1.8E-04 | 1.6E-10 |
| 4  | 3.8E-10 | 1.6E-10 | 1.3E-10 | 6.1E-11 | 7.6E-07 | 1.8E-03 | 4.2E-10 |
| 5  | 2.5E-09 | 3.2E-10 | 2.9E-10 | 1.3E-10 | 7.6E-06 | 1.8E-02 | 2.9E-09 |
| 6  | 2.4E-08 | 1.9E-09 | 1.8E-09 | 8.0E-10 | 7.6E-05 | 1.8E-01 | 2.8E-08 |

**6) 3, 36 "2.35E4 ncps/ppb". Ncps is not defined until the subsequent paragraph; consider re-ordering or inserting 'as defined below'. I suggest also reporting here the raw sensitivity in cps/ppb as this gives a more directly interpretable measure of the instrumental response / LOD.**

Definition issue has been fixed. Regarding your suggestion: The CIMS data reported here are normalized to reagent ion (acetate) signal, to ensure that any observed changes in signal are not due to changes in reagent ion signal.

**7) 4, 5: misplaced comma at beginning of line**

Fixed.

**8) 5, 13 "biogenic emissions typically cease during the night". Should specify here that this is the case for light-dependent emissions but not for solely temperature-dependent emissions.**

This sentence has been rewritten as the following: *"Further, the near-surface source persists through both day and night, while biogenic light-dependent emissions typically cease during the night when stomata are closed and photosynthesis has stopped."*

**9) 3, 14 "PISA" not defined**

Fixed.

[revised manuscript text omitted]

$$[X]_{aq} = H_x P_x \left(1 + \frac{K_a}{[H^+]}\right)$$

where K$_a$ is the acid dissociation equilibrium constant for a given acid (Levanov et al., 2017; Fischer and Warneck, 1991; Borduas et al., 2016; Smith and Martell, 2004), and [H$^+$] is the aqueous concentration of hydronium ion. Combining aqueous-phase moles of a given acid with the ideal gas law, and meteorological data from the site yields a total loss of said acid from the gas-phase through partitioning. Total loss of each acid calculated at various atmospherically-relevant pH values are reported below. This estimation is limited in that it neglects the effects of other dissolved ions on solubility, though we would not expect a change of several orders of magnitude by accounting for these effects.

| | Loss via aqueous partitioning (ppbv) | | | | | | |
|---|---|---|---|---|---|---|---|
| pH | Formic | Propionic | Butyric | Valeric | Pyruvic | Nitric | Isocyanic |
| 2 | 1.4E-10 | 1.4E-10 | 1.1E-10 | 5.4E-11 | 1.5E-08 | 1.8E-05 | 1.4E-10 |
| 3 | 1.6E-10 | 1.4E-10 | 1.2E-10 | 5.4E-11 | 8.3E-08 | 1.8E-04 | 1.6E-10 |
| 4 | 3.8E-10 | 1.6E-10 | 1.3E-10 | 6.1E-11 | 7.6E-07 | 1.8E-03 | 4.2E-10 |
| 5 | 2.5E-09 | 3.2E-10 | 2.9E-10 | 1.3E-10 | 7.6E-06 | 1.8E-02 | 2.9E-09 |
| 6 | 2.4E-08 | 1.9E-09 | 1.8E-09 | 8.0E-10 | 7.6E-05 | 1.8E-01 | 2.8E-08 |

Aqueous-phase partitioning was evaluated as a potential sink for gas-phase acids by using Henry's Law:

$$H_x = \frac{[X]_{aq}}{P_x}$$

where $H_x$ is the Henry's Law constant for a given gas-phase acid, and $[X]_{aq}$ and $P_x$ are the aqueous concentration and partial pressure of said acid species, respectively. $P_x$ was calculated by gas-phase acid mixing ratio data, as well as meteorological data collected during the campaign. Moles of a given acid in the aqueous-phase was determined by $[X]_{aq}$ and ambient liquid water concentration (LWC). LWC in the Front Range during the summer is estimated to be around 1 $\mu$g m$^{-3}$, based on continental estimates of LWC reported by Carlton and Turpin (2013). Combining aqueous-phase moles of a given acid with the ideal gas law, and meteorological data from the site yields a total loss of said acid from the gas-phase through partitioning. This estimation is limited in that it does not account for the effects of pH or other dissolved ions of a given acid's solubility, but we would not expect a change of several orders of magnitude by accounting for these effects.

**Supplemental References**

Borduas, N., Place, B., Wentworth, G., Abbatt, J., and Murphy, J.: Solubility and reactivity of HNCO in water: insights into HNCO's fate in the atmosphere, 16, 703-714, 2016.

Carlton, A. G., and Turpin, B. J.: Particle partitioning potential of organic compounds is highest in the Eastern US and driven by anthropogenic water, Atmos. Chem. Phys., 13, 10203-10214, doi:10.5194/acp-13-10203-2013, 2013.

Fischer, M., and Warneck, P.: The dissociation constant of pyruvic acid: determination by spectrophotometric measurements, 95, 523-527, 1991.

Levanov, A., Isaikina, O. Y., and Lunin, V.: Dissociation constant of nitric acid, 91, 1221-1228, 2017.

Li, Y., Schwandner, F. M., Sewell, H. J., Zivkovich, A., Tigges, M., Raja, S., Holcomb, S., Molenar, J. V., Sherman, L., Archuleta, C., Lee, T., and Collett, J. L.: Observations of ammonia, nitric acid, and fine particles in a rural gas production region, Atmos. Environ., 83, 80-89, doi:10.1016/j.atmosenv.2013.10.007, 2014.

Seinfeld, J. H., and Pandis, S. N.: Atmospheric Chemistry and Physics, 1 ed., Wiley-Interscience, Canada, 1998.

Smith, R. M., and Martell, A. E.: NIST Standard Reference Database 46, in, 2004.